

# Decomposition technique for contributions to groundwater heads from inside and outside of an arbitrary boundary: Application to Guantao County, North China Plain

Ning Li[1], Wolfgang Kinzelbach[1], Haitao Li[2], Wenpeng Li[2], Fei Chen[3]

[1]Institute of Environmental Engineering, ETHZ, Zurich, 8093, Switzerland
[2]China Institute of Geo-Environment Monitoring, Beijing, 100081, China
[3]General Institute of Water Resources and Hydro-power Planning and Design, Beijing, 100081, China

*Correspondence to*: Ning Li (ning.li@ifu.baug.ethz.ch)

**Abstract.** To assess the efficiency of groundwater management of an administrative unit, we propose to decompose the groundwater head changes within an administrative unit into inside and outside contributions by using numerical models. Guantao County of Hebei Province, China, serves as an example to demonstrate the decomposition technique. The groundwater flow model of Guantao was constructed using observed heads as prescribed head boundary conditions. The model was coupled with Hydrus 1D, to calculate the groundwater recharge distribution in time reflecting the delay and damping effects of the soil column on seepage at the surface. The model was calibrated by adjusting parameters such as hydraulic conductivities, recharge infiltration ratios and specific yields. The calibrated parameters are then used in a large model with a boundary at large distance from Guantao administrative boundary to determine the groundwater head changes due to inside drivers. The differences of the two models on the Guantao boundary serve as the specified head values on the boundary for a small scale model, which is used to calculate the groundwater head imposed by outside drivers. To eliminate inconsistencies caused by the different types of boundary conditions of large and small models, the groundwater head changes due to inside drivers must be updated. The results indicate that the groundwater head changes in the centre and south of Guantao County are influenced equally by both inside and outside contributions, while in the north outside contributions have the stronger impact. The sensitivity analysis shows that the groundwater head changes and their decomposition are much more sensitive to infiltration ratios than to the aquifer parameters. The parameters within Guantao have a certain influence on the net groundwater head changes while the parameters outside of Guantao have only an influence on the decomposition.

## 1 Introduction

The natural shallow groundwater flow field is usually determined by infiltration from precipitation and streams, by discharge to streams or springs and by phreatic evapotranspiration. With the development of human activities, the natural groundwater levels are modified by land use change, crop planting and irrigation, groundwater pumping and more. Over the last few





decades, the groundwater resources world-wide have been intensively used for household, industry and agriculture purposes (Zektser and Everett, 2004). It is estimated that of the approximately 1000 km3 of groundwater pumped worldwide per year about one quarter is not replenished by recharge anymore (WWAP, 2012; Wada et al., 2012). The major user of groundwater is irrigated agriculture. Especially in the North China Plain (NCP), groundwater became the major water resource for

agriculture accounting for more than 70% of the total water supply (Wu, 2013). The continuous unsustainable abstraction of groundwater caused a dramatic groundwater level decline:  by 18 m in the piedmont region in 25 years (Chen et al., 2003), by 5-50 m in the shallow aquifer and by 10-100 m in the deep aquifer over the last 50 years (Wu, 2013). This severe groundwater level decline causes land subsidence, drying up of the lower reaches of rivers and streams, salt-water intrusion at the Hebei Coast, and municipal water shortages (Wu, 2013; Liu et al., 2001; Han et al., 2011). It also increases the cost of

pumping groundwater. To support decision makers to put forward rational water management strategies in such areas and alleviate groundwater decline or even reach some recovery of groundwater levels, it is useful to separate impacts on groundwater head changes caused by different drivers. Hydrological models have increasingly been applied in assessing the influences on groundwater levels due to individual drivers such as precipitation, changes in cropping system, and irrigation practice, water imports and others (Shu et al., 2012; Cao et al.2013; Li, 2013; Iwasaki et al., 2014 ). For a small region

within a larger aquifer system it might be of interest to identify the influences of drivers within and outside of the region.

To build a groundwater model of an administrative unit is difficult as usually natural model boundaries do not coincide with political boundaries. Yet, we often want to assess the efficiency of groundwater management of an administrative unit. The groundwater heads within an administrative unit are not only influenced by the groundwater head changes induced by inside

drivers but also by the groundwater head changes induced by outside drivers. Monitoring groundwater levels within the administrative unit often cannot directly and faithfully reflect the effects of water management measures implemented in that unit. In other words, the monitored groundwater head changes are caused by both inside and outside contributions. Therefore, we propose to decompose the groundwater head changes over a given time period within an administrative unit into inside and outside contributions to assess the efficiency of the unit's management of groundwater resources. To do so

three groundwater models are required:

(1) a model of the administrative unit with the political boundary implemented as a prescribed head boundary using observed heads,

(2) a larger model containing this model, which computes only the propagation of inside changes to the outside and which may extend to physical model boundaries and

(3) a model of the administrative unit with prescribed heads obtained by subtraction from the previous two models on the political boundary, which is used to determine the groundwater head contributions controlled by outside drivers only.

Guantao County is located in the NCP and delimited by a political administrative boundary. To test aquifer rehabilitation measures, Guantao is chosen as a pilot region for research. When applying remedial measures in Guantao, e.g. by decreasing



water abstraction from the aquifer via suitable measures, the groundwater level in Guantao County could increase, decrease or show no change with time depending on the superimposed influence of the outside groundwater flow field. This impedes the assessment of the effects and the efficiency of Guantao water management strategies. The solution to fair assessment lies in decomposing the groundwater head changes in Guantao into two parts: one part is the head change due to the influence of

recharge and discharge in Guantao (inside contribution), and the other is the head change contribution determined by outside drivers, i.e. the groundwater flow field without considering source and sink terms in Guantao (outside contribution).

Numerous studies were carried out in the NCP to understand the large scale groundwater dynamics employing MODFLOW (Lu et al., 2011; Zhou et al., 2012; Qin et al., 2013). For local management purposes these models are far too crude. But

groundwater simulations in smaller administrative units of the NCP are quite rare up to now. In our study, a groundwater flow model for Guantao is constructed using MODFLOW USG under the pre-and postprocessor Visual MODFLOW Flex 2014.1 (VMF) developed by Waterloo Hydrogeologic (Panday et al., 2013).

The objective of this research is to take Guantao as a case study to demonstrate how to decompose the groundwater head

changes into inside and outside contributions using numerical models. First the general background and available data are introduced. Then the mathematical method of decomposing the groundwater head changes is described in detail. Following that, a numerical model for the Guantao aquifer system is developed and calibrated under both steady state and time-varying conditions. Then, both a large scale and a small scale numerical model each are developed to separately determine the groundwater head changes due to inside and outside contributions. Finally, the sensitivity of the groundwater head changes

and their decomposition with respect to parameters and local management measures are discussed.

## 2 Study area and data

### 2.1 Study area

The NCP is bordered by the Tai Hang Mountains in the west, the Yellow River in the south, the Bohai Sea in the east and the Yan Shan Mountains in the North and has an area of altogether 140,000 $km^2$ (Fig.1). Guantao County is located in the

southern NCP, between longitudes 115°06'-115°28' and latitudes 26°27'-36°47', covering an area of 456 $km^2$ (Fig. 1). There are 8 townships and 277 villages under the county's administration with a population of 340'000 (GSB, 2011). From the piedmont in the west to the coast in the east, the NCP is divided into four geomorphological units with Guantao lying on the alluvial plain zone (Wu et al., 1996). The topography of Guantao is quite flat, with an elevation of 45 m above sea level (a.s.l.) in the southwest, which is only slightly higher than the value of 36 m a.s.l. in the northeast.

**Figure 1: Schematic map of the NCP (a) and administrative map of Guantao (b)**




Guantao has the characteristics of a warm continental monsoon climate: hot and rainy in summer, and cold and dry in winter with an average annual precipitation of 532 mm, an average annual potential evaporation of 1516 mm and an annual mean temperature of 13.4°C (GEF project, 2009). About 80% of the annual precipitation occurs between June and September. Weiyun River is a seasonal river, which dries up outside of the monsoon season. It flows along Guantao's east boundary

between June and October with a flow of $0.4 \times 10^9$ m³/a averaged over the period 2001-2012 (data collected by GIWP). Weixi channel is the main irrigation channel diverting surface water from Weiyun River to crop lands. The total cultivated area is around 300 km² (data from 2013) and the main crops are winter wheat, summer corn and cotton. Winter wheat is sown in October and harvested in June of the following year. After wheat harvest, the summer corn is immediately planted on the same area and harvested in September. The cotton growing period is between April and September. The annual

diversion in the channel is only around $3.1 \times 10^6$ m³/a (average between 2001 and 2012, data collected by GIWP). During the monsoon season, less irrigation is needed because the concentrated precipitation is sufficient to satisfy crop needs most of the time. The non-monsoon season coincides with the growing season of winter wheat, which means irrigation with groundwater is the only way to support winter wheat growth.

The Quaternary aquifer of Guantao consists of fine sand layers interbedded with clay or silt aquitard layers. Vertically it is divided into shallow, middle and deep layers according to different deposits. Based on previous studies and the recent hydrogeological investigation by CNACG (2015), the middle layer mainly consists of clay with an average thickness of around 120 m. It is saline with total dissolved solids reaching 10,000 g/m³ in some parts (GEF project, 2009). The shallow layer is exclusively used for irrigation water supply while the deep layer is mainly used for domestic and industrial water

supply. Still, the deep layer has to provide some irrigation water too due to locally excessive salinity in the shallow aquifer. The groundwater level differences between deep and shallow aquifers are huge and the main chemical groundwater characteristics in both layers are totally different (CNACG, 2015). Hence, the middle layer is regarded an efficient aquitard which practically does not allow a direct hydraulic connection between deep and shallow layers. The shallow aquifer is recharged by precipitation, irrigation backflow and channel and river seepage. It is discharged through pumping wells. The

deep layer receives only little recharge from upstream at the piedmont, so groundwater levels have been decreasing since pumping started. Long-term intensive groundwater pumping caused many environmental problems such as groundwater depletion, land subsidence, and sea water intrusion. Stopping all abstraction from the deep layer is the only way to let groundwater levels in the deep aquifer recover. The political decision has been taken to stop almost all pumping from the deep aquifer and replace this water by water from the South North water transfer scheme. By 2018 the replacement of

domestic and industrial use water was concluded while agriculture is still pumping from the deep aquifer. In this study, the shallow aquifer is the main concern and the modelling effort comprises only this layer.





## 2.2 Data

All data required to calibrate a numerical groundwater model for the time between 2003 and 2012 are available. Groundwater heads (provided by GIWP and Handan DWR) are observed at 3 different frequencies. There are 4 observation points with monthly measurements, 11 observation points with 2 data records each year and 29 observation points with 4

measurements each year. Daily precipitation data has been purchased from Guantao Meteorological Bureau (yearly data of precipitation between March and October shown in Fig. 2). Annual channel diversions for irrigation and monthly runoff of Weiyun River as well as officially recorded annual pumping rates have been collected from the Water Resources Bulletin of Guantao for the period from 2003 to 2012 (Fig. 2). The pumping rate for irrigation is highly dependent on precipitation in the NCP. Pumping rates are generally less in years with higher precipitation. Hence, the data point for 2003 showing a

combination of higher precipitation with a larger pumping rate is questionable. The sudden significant decrease in the reported pumping rate after 2006 is also questionable as there were hardly any changes in cropping area, crop types and irrigation methods. Therefore, the pumping rates for these years will be adjusted during calibration of the numerical model.

**Figure 2: Time series of the annual pumping rates in the shallow aquifer shown together with annual precipitation (solid line:**

**reported data, dashed line: values adjusted during calibration).**

## 3 Methods

### 3.1 Equations

Groundwater heads in Guantao ( $h_{\text{Guantao}}(t, x, y)$ ) result from the superposition of groundwater head changes due to inside drivers ( $\Delta h_{\text{Guantao}}(t, x, y)$ : inside drivers) and groundwater heads determined by outside drivers ( $h_{\text{out}}(t, x, y)$ : outside

drivers). The decomposition approach is written as follows,

$$h_{\text{Guantao}}(t, x, y) = \Delta h_{\text{Guantao}}(t, x, y) + h_{\text{out}}(t, x, y) \tag{1}$$

The governing equation for the transient groundwater flow in a phreatic aquifer can be described using a linear partial differential equation (PDE) with a time-independent transmissivity provided the head changes in a predefined period are relatively small compared to the aquifer thickness.

$$\mu \frac{\partial h}{\partial t} = T \frac{\partial^2 h}{\partial x^2} + T \frac{\partial^2 h}{\partial y^2} + q^{\text{in}} - q^{\text{out}} \tag{2}$$

$h$ refers to the groundwater head (m), $T$ is the transmissivity (m²/s), $\mu$ is the specific yield (-), t is the time (s), $q_{in} \text{ and } q_{out}$ are source and sink terms respectively (m/s).

The groundwater heads in Guantao can be determined by the governing equation below together with a (measured) specified

head boundary along the administrative boundary.



$$\mu \frac{\partial h_{\text{Guantao}}}{\partial t} = T \frac{\partial^2 h_{\text{Guantao}}}{\partial x^2} + T \frac{\partial^2 h_{\text{Guantao}}}{\partial y^2} + q_{\text{Guantao}}^{\text{in}} - q_{\text{Guantao}}^{\text{out}} \tag{3}$$

$q_{\text{Guantao}}^{\text{in}}$ includes not only the anthropogenic sources (irrigation backflow and channel infiltration) but also the natural recharge by precipitation. River seepage is included in the prescribed head boundary as the boundary coincides with the Weiyun River.

Accordingly, the PDE to determine the groundwater head changes due to inside drivers is as follows,

$$\mu \frac{\partial \Delta h_{\text{Guantao}}}{\partial t} = T \frac{\partial^2 \Delta h_{\text{Guantao}}}{\partial x^2} + T \frac{\partial^2 \Delta h_{\text{Guantao}}}{\partial y^2} + q_{\text{Guantao}}^{\text{in}} - q_{\text{Guantao}}^{\text{out}} \tag{4}$$

To solve Eq. (4), apart from the knowledge of sources and sinks in Guantao, a proper choice of the model boundary is also important. The ideal boundary is chosen at a large distance from Guantao County administrative boundary to make sure that

the influence of sources and sinks within Guantao on this distant boundary can be assumed to be equal to zero.

The groundwater heads in Guantao caused by outside drivers only can be obtained by numerically solving the following flow equation,

$$\mu \frac{\partial h_{\text{out}}}{\partial t} = T \frac{\partial^2 h_{\text{out}}}{\partial x^2} + T \frac{\partial^2 h_{\text{out}}}{\partial y^2} + q_{\text{outside}}^{\text{in}} - q_{\text{outside}}^{\text{out}} \tag{5}$$

$q_{\text{outside}}^{\text{in}}/q_{\text{outside}}^{\text{out}}$ are the source/sink terms outside of Guantao. The model used to solve Eq. (5) is only established for Guantao area, where the $q_{\text{outside}}^{\text{in}}/q_{\text{outside}}^{\text{out}}$ terms are equal to zero. A specified head boundary condition is defined on the administrative boundary, which can be obtained as the difference between the boundary condition of Eq. (3) and the solution of Eq. (4) on the county boundary. Anyway, when two variables in Eq. (1) are known, the third variable is determined as well.

As mentioned above, to decompose the groundwater heads in Guantao, three numerical models are used in this research: The first is a small-scale numerical model within Guantao's administrative boundary to calculate $h_{\text{Guantao}}(t, x, y)$ (Guantao flow model). It uses measured heads on its (specified head) boundary, which coincides with the county border. The second is a larger scale numerical model extending to distant natural boundaries to calculate $\Delta h_{\text{Guantao}}(t, x, y)$ (large flow model). Its

boundary conditions are zero head change induced by Guantao sources and sinks. The third one is again a small-scale numerical model within Guantao's administrative boundary to calculate $h_{\text{out}}(t, x, y)$ ( $h_{\text{out}}$ flow model in Guantao). The specified heads on the boundary are now determined by sources and sinks outside of Guantao only. They can be obtained from the measured heads by subtracting the head changes obtained from the large flow model on the county boundary. The analysis of the decomposed groundwater heads can be carried out at any time and any location within Guantao County. For a

predefined time period from $t$ to $t + \Delta t$ Eq. (1) can be restated as Eq. (6), which says that the groundwater head changes




over that period ($\delta h_{\text{Guantao}}(\Delta t, x, y)$) can also be decomposed into head changes due to inside contributions ($\delta \Delta h_{\text{Guantao}}(\Delta t, x, y)$) and outside contributions ($\delta h_{\text{out}}(\Delta t, x, y)$).

$$\delta h_{\text{Guantao}}(\Delta t, x, y) = \delta \Delta h_{\text{Guantao}}(\Delta t, x, y) + \delta h_{\text{out}}(\Delta t, x, y) \qquad (6)$$

In which

$$\delta h_{\text{Guantao}}(\Delta t, x, y) = h_{\text{Guantao}}(t + \Delta t, x, y) - h_{\text{Guantao}} \qquad (7)$$

$$\delta \Delta h_{\text{Guantao}}(\Delta t, x, y) = \Delta h_{\text{Guantao}}(t + \Delta t, x, y) - \Delta h_{\text{Guantao}}(t, x, y) \qquad (8)$$

$$\delta h_{\text{out}}(\Delta t, x, y) = h_{\text{out}}(t + \Delta t, x, y) - h_{\text{out}}(t, x, y) \qquad (9)$$

for any point ($x$, $y$) within Guantao borders. As we are mainly interested in changes of heads both due to inside and outside contributions, we will in the following use the formulations of Eq. (6) through Eq. (9). $\Delta t$ refers to the whole simulation period of 10 years.

### 3.2 Model description

#### 3.2.1 Guantao flow model

The Guantao model boundary is the administrative boundary of Guantao, which is conceptualized as a specified head boundary condition. All available groundwater head observations close to the boundary are made use of defining the boundary heads by interpolation. The bottom of the aquifer (lower boundary of the model) is defined by the boundary between the shallow aquifer and the aquitard. It is interpolated from the well logs of the hydrological investigation (CNACG, 2015) and considered impervious.

Precipitation is the main recharge to groundwater in Guantao. Other natural recharge terms include river infiltration of Weiyun River on the east boundary and infiltration from the Weixi channel passing through Guantao from South to North. The Weiyun River infiltration is not explicitly described in the numerical model but implicitly considered in the specified head boundary condition. The Weixi channel infiltration is simulated using the river package of Modflow. The river level is assumed to be 3 m higher than the deepest river channel bottom level. Guantao is an agricultural county. Apart from the main Weixi channel, there are still numerous smaller sub-channels connected to the Weixi channel to divert and store surface water. Infiltration from these smaller sub-channels is considered part of the areal recharge from irrigation and simulated as a contribution to the recharge package.

Groundwater is the main source for irrigation. This is very common in the NCP (Hu et al., 2010; Zhang et al., 2010). In Guantao, due to lack of surface water, groundwater accounts for over 95% of total applied irrigation water according to data



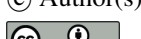

between 2001 and 2011. Nowadays the traditional conveyance of irrigation water by canals and ditches is replaced by pipe irrigation to increase the conveyance efficiency from wells to the cropland. On the field, ridge irrigation and furrow irrigation are still the primary methods for both groundwater and surface water irrigation. Hence, the irrigation backflow is another important source of groundwater recharge. In the study, the recharge contributions by precipitation, irrigation

backflow and canal seepage were added up and implemented in the recharge package. The distribution of agricultural area is obtained from satellite remote sensing images of 2014, by mapping NDVI in winter and spring (provided by Dr. Haijing Wang, hydrosolutions Ltd.). The total agricultural area changes only slightly between 2002 and 2013 according to the statistical year books (GSB, 2002-2013). Hence, the agricultural area is assumed to be constant over the modelled time span.

The depth to groundwater is around 20 m according to the available groundwater head observations, so there is no phreatic evaporation from the aquifer. The discharge is exclusively due to pumping wells extracting groundwater from the aquifer. The water abstraction by pumping is implemented in the well package and the spatial distribution is chosen according to the well locations obtained from the general investigation in 2011 (local communication). The pumping rate of the individual well was determined as a constant fraction of the estimated total annual groundwater abstraction (see section below).

The 2-dimensional groundwater flow model is constructed using VMF-USG, the unstructured grid version of USGS-Modflow. The spatial discretization consists of 141,379 cells (Fig. 3). The cells are refined around pumping wells. The top elevation is interpolated from SRTM data downloaded from the website (http://srtm.csi.cgiar.org). The thickness of the aquifer is between 55 m and 96 m. The hydraulic conductivity ($K$) is distributed by zonation in the model domain. Four

zones are defined based on pumping tests carried out by CNACG (2015) (Fig. 4). Fourteen groundwater head observations are used to define the fixed head boundary. They divide the boundary into segments with observations at both ends of a segment. The cell values within a segment are linearly interpolated from the observations on both ends. Missing observation data in some stress periods are linearly interpolated from two close data records of the time series. The adjusted annual groundwater abstraction is assigned to monthly stress periods within a year based on irrigation water application. Irrigation is

mainly applied outside of the monsoon season. According to local information, winter wheat is primarily irrigated in October, March and April, summer maize in May and June and cotton in April and May. So the groundwater abstraction is set to zero in all other monthly stress periods.

**Figure 3: Discretization map of Guantao model**

**Figure 4: Zonation of hydraulic conductivities and recharge infiltration ratios**

Recharge is the most difficult input to quantify in the simulation. Some researchers suggest that it is a non-linear function of the water input at the surface (Kendy, 2003; Kendy, 2004; Tan, 2014; Min, 2015). To simplify the process, the recharge





infiltration ratio (*R*) is divided into three zones according to the soil map (CNACG, 2015) (Fig. 4). For each zone, the recharge fluxes from precipitation and irrigation are estimated as constant fractions of annual precipitation and irrigation amounts as suggested in other literature on the groundwater system in the NCP (Jia et al., 2002; Liu et al., 2008; Shao et al., 2013). These recharge infiltration ratios will be calibrated in the steady state model.

However, for a transient calculation, the temporal distribution of recharge also has to be represented adequately to describe its influence on groundwater level dynamics correctly. In order to better understand the time-lag of the groundwater level response to the water input at ground surface, the groundwater movement in the unsaturated soil zone was simulated using Hydrus 1D (Simunek et al., 2013) in this study. The soil column is modelled from the ground surface to a depth of 20 m,

10 which is comparable to the yearly averaged depth to groundwater table in Guantao over the last 10 years. Lu et al. (2011) studied the groundwater recharge at five representative sites in the NCP (two in the piedmont plain, two in the alluvial plain and one in the coastal plain) using Hydrus. Guantao is located in the alluvial plain of the NCP with soils mainly consisting of silt, clay and silty clay. Since no field sampling, monitoring or experiments in the soil zone were carried out within the project, the soil material is assumed to be uniformly distributed in the column. The soil properties are adopted from the two

15 representative sites in the alluvial plain cited in Lu et al. (2011). The van Genuchten parameters chosen as input into Hydrus are the average values from Lu's study (2011): $\theta_{residual}$= 0.08, $\theta_{saturated}$ = 0.39, $\alpha$ = 0.54 (1/m); n = 1.98; and $k_s$ = 0.65 (m/d). The simulated soil column is vertically discretized into 100 layers with a spacing of 0.2 m. A self-adjusting numerical time discretization was adopted in the simulation with a minimum time step of 0.0001 days and a maximum time step of 1 day. The flexible time discretization in the simulation ensured convergence of the numerical calculation.

The upper boundary is implemented as an atmospheric boundary to the surface layer while the free drainage condition is implemented at the lower boundary. The water input for the groundwater surface equals precipitation plus irrigation minus evaporation and evapotranspiration, which in fact is the seepage rate turning to groundwater recharge. The initial water content distribution is obtained by running the simulation periodically for 10 years to get a relatively steady distribution.

The aim of the model calibration is to minimize the residuals between the observed and computed head values by adjusting model parameters. We start out with a steady state model. The average behaviour in the period from 2003 to 2011 was used as a quasi-steady state. This is feasible, as the average head over all observations at the end of 2002 is almost identical to that at the end of 2011. That means that on average, abstractions must have been balanced by recharge over that period. All

30 inputs required in this case are average data sets between 2003 and 2011. The steady state model calibration was accomplished using the automated parameter estimation code (PEST) (Doherty, 2003). The transient flow model between 2003 and 2012 is calibrated manually, as only the specific yield values have to be adjusted. The groundwater flow field obtained from the steady state model is used as the initial condition of the transient model. The transient model time span was divided into monthly stress periods. There are in total 120 stress periods. The whole calibration process is shown in Fig.



5. Initially, the officially reported (but estimated) time series of the pumping rate and inferred recharge ratios are input to Hydrus 1D to obtain the average groundwater recharge between 2003 and 2011. Hydrus 1D is run for each township using different seepage water input values according to the collected data. Then the parameter estimation model PEST is applied to calibrate parameters ($K$ and $R$). The distribution of groundwater recharge from Hydrus is input to the transient model and

specific yield values are calibrated manually. The specific yield influences both the amplitudes of heads within the year and their rate of change over the longer term. When the yearly averaged calculated head value at all observation wells is far away from the corresponding observed value, the recorded pumping rate in that year is adjusted first. In this application, the biggest deviation between computed and measured heads is controlled within 1 m. The whole calibration process is implemented iteratively to converge to the best fit to observations.

**Figure 5: Flow chart of the calibration process of the Guantao model**

After the model calibration, several statistical indices were used to assess the results of the numerical model. The correlation coefficient R-square is used to measure the correlation between modelled and observed groundwater heads in the steady state

case. Root mean square error ($E_{RMS}$) is used to evaluate how closely modelled groundwater heads are fitting observed data time series and Mean Absolute Error ($E_{MA}$) is used to evaluate the average magnitude of errors we can expect from modelled groundwater heads. Both $E_{RMS}$ and $E_{MA}$ are used to evaluate the model performance on average. They are defined as follows:

$$E_{RMS} = \sqrt{\frac{\sum_{i=1}^{m}\sum_{t=1}^{n}(Y_{i,t}-O_{i,t})^2}{m*n}} \qquad (10)$$

$$E_{ME} = \frac{\sum_{i=1}^{m}\sum_{t=1}^{n}|Y_{i,t}-O_{i,t}|}{m*n} \qquad (11)$$

where $Y_{i,t}$ is the $i^{th}$ modelled groundwater head at time $t$; $O_{i,t}$ is the corresponding observed value at the same time step; $m$ is the number of observation wells; $n$ is the number of time steps.

The zonal values of hydraulic conductivities and recharge infiltration ratios obtained from PEST in the calibration of the

steady state model are presented in Table 1. The values of hydraulic conductivities indicate a decreasing trend from south to north. The relatively less permeable zone between $K$1 and $K$3 is also consistent with the local hydrological knowledge. The distribution of the recharge infiltration ratios shows that the values tend to decrease from east to west. Tan (2014) shows that in the NCP the average recharge ratio from tracer experiments is 0.16 ranging from 0.11 to 0.21. The average value from numerical simulations is 0.14 ranging from 0.08 to 0.2. Kendy (2003) points out that the recharge ratio obtained from soil

water balances at 16 sites in the NCP ranges from 0.09 to 0.24. Kendy (2004) concluded that the recharge ratio in the NCP varied between 0.2 and 0.6 in numerical model simulations. Min (2015) proved that the recharge ratio varied between 0.07





and 0.53 in a 30years' Hydrus simulation. Lu (2011) concluded that the recharge ratio is 0.18 in the alluvial plain according to Hydrus simulations. The former studies reported a wide range of values for the recharge infiltration ratio in the NCP with an average value around 0.23. This further proves that the calibrated recharge ratios in our study are reasonable and acceptable. During the auto-calibration process, $R3$ tends to be extremely small with a value even less than 1%. To avoid

this unrealistic number, $R3$ is assumed to be fixed in the calibration at a value of 0.09, which is the average of the minimum recharge ratios obtained from the previous studies in the NCP.

The sensitivities of different parameters to observed heads are shown in Table 1, which indicates that compared to hydraulic conductivities the recharge infiltration ratios have a much larger influence on groundwater heads. The sensitivity of

hydraulic conductivity is higher in the south than in the north. The most sensitive recharge infiltration ratio is found in the east. The correlation coefficients among different parameters are presented in Table 2. Correlation coefficients with a modulus above 0.5 indicate that there is a non-negligible correlation among the respective parameters. The highest (anti-) correlation coefficient between two zonal hydraulic conductivities is -0.58 (between $K2$ and $K3$). This shows that their relative values cannot be uniquely determined by calibration. The recharge ratios $R1$ and $R2$ are also correlated with a

correlation coefficient of -0.57, which means that the sum of the recharge in zones 1 and 2 is more important than the individual values. The R-square of the comparison between simulated groundwater heads and observations is around 0.83, which implies that 83% of the variance of observations could be explained by the numerical model. Although the fit to the observed data is very good considering the data situation, it has to be noted that the result is still uncertain. An increased pumping rate with a simultaneously increased recharge ratio will lead to a similar result. For a unique result, accurate

pumping rates are required.

Table 1. The automatically calibrated values of hydraulic conductivities and recharge infiltration ratios and their respective sensitivity to observed heads

| Parameter | Values (m/d or-) | 95% percent confidence limits | | Sensitivity |
| --- | --- | --- | --- | --- |
| | | lower limit | upper limit | |
| $K1$ | 11.7 | 7.9 | 17.3 | 0.42 |
| $K2$ | 4 | 2.4 | 6.6 | 0.50 |
| $K3$ | 8 | 3 | 21 | 0.20 |
| $K4$ | 1.7 | 0.8 | 3.6 | 0.26 |
| $R1$ | 0.16 | 0.12 | 0.2 | 1.35 |
| $R2$ | 0.21 | 0.19 | 0.22 | 4.13 |

Table 2. Correlation coefficients among different parameters




|     | $K1$ | $K2$ | $K3$ | $K4$ | $R1$ | $R2$ |
|-----|------|------|------|------|------|------|
| $K1$ | 1.00 | -0.28 | 0.02 | 0.10 | -0.05 | -0.06 |
| $K2$ |      | 1.00 | -0.58 | 0.36 | -0.49 | -0.11 |
| $K3$ |      |      | 1.00 | -0.10 | 0.13 | 0.31 |
| $K4$ |      |      |      | 1.00 | -0.36 | -0.25 |
| $R1$ |      |      |      |      | 1.00 | -0.57 |
| $R2$ |      |      |      |      |      | 1.00 |

The recharge fluxes at different soil depths for one of the eight townships are shown in Fig. 6. They capture the important characteristics of the soil water flux at different depths. The response time becomes longer with increasing depth to groundwater while the peak flux becomes smaller with a more even distribution over the year. Although the water input on
the soil surface is quite different from year to year, few peaks are observed when the soil depth is larger than 10 m, due to damping.

The peaks vanish faster with increasing soil depth in dry years than in wet years. Despite the averaging, the bottom flux distribution over a year is different from year to year. Even at 20 m depth the biggest difference between the highest and
lowest flux is still around 30% of the lowest flux. The groundwater recharge is a complex nonlinear process which mixes all the irrigation and precipitation events occurring at different times. Through the analysis of results from Hydrus, it is obvious that the best way to accurately describe the groundwater recharge in the groundwater model is to couple the model of the unsaturated zone with the groundwater model. The recharge fluxes for other townships demonstrate a similar distribution.

**Figure 6: Calculated monthly recharge flux at different soil depths**

The specific yield has the same zonation as the hydraulic conductivity. Four zones of specific yield, denoted by $S_y1$, $S_y2$, $S_y3$ and $S_y4$, are calibrated manually to reflect the correct head amplitudes. The specific yields decrease from south to north with the respective values of 0.05 ($S_y1$), 0.03 ($S_y2$), 0.02 ($S_y3$) and 0.01 ($S_y3$). The adjusted pumping rates are shown in Fig. 2.
They are consistent with the common experience that less groundwater is pumped in wet years while the water demand increases in dry years.

The goodness of fit between modelled and observed groundwater heads is presented in Fig. 7. Some computed heads underestimate observations and others overestimate observations. There is no general bias. The seasonal pattern of
groundwater level dynamics is reproduced. The groundwater level starts to decrease from March due to irrigation and reaches its lowest value in June or July. From June or July on, groundwater levels start to recover and increase in response to termination of pumping and the delayed recharge by precipitation and irrigation. The large deviations between observations



and computation during the first year are caused by inconsistent initial groundwater heads, which are "forgotten" after about 1 year. Remember that the steady state solution taken as initial head distribution corresponds more to the average situation over the 10 year period than to the initial distribution in 2003. The $E_{\mathrm{RMS}}$ and $E_{\mathrm{AM}}$ between observations and simulated groundwater heads are 1.62 m and 1.18 m respectively. The relative small values of $E_{\mathrm{RMS}}$ and $E_{\mathrm{AM}}$ imply that there is no big

variance in the individual errors between observations and calculations. These errors are still acceptable if compared to the groundwater level variations in Guantao. Although the fit is very good due to the adjustment of pumping rates, it has to be kept in mind that there are many ways to do this adjustment and final results are uncertain. It is clear that the calibrated model used here is not unique due to the resulting correlation among parameters and pumping rates. But this calibrated model is a completely consistent model considering the iterative calibration process. Due to the constraints on parameters,

the coefficients of the covariance matrix stay relatively small (largest matrix element in table 2 is 0.58 in modulus). Therefore, the calibrated model can be regarded as a representative realization of reality and used for the further analysis. All aspects of uncertainty will be treated in a real-time model version, in which uncertainty is taken into account by ensemble techniques (paper in preparation).

**Figure 7: Calculated and observed monthly time series of groundwater heads at four available long-term observation wells**

### 3.2.2 Large flow model

Sources and sinks in the large flow model are conceptualized similarly as in the Guantao model. The infiltration from precipitation and infiltration from applied irrigation water were calculated with Hydrus 1d and combined in the groundwater recharge package. The Weixi channel infiltration is simulated as defined flux boundary instead of using the river package.

The infiltration flux is extracted from the Guantao model and then assigned to corresponding cells in the large model. The groundwater discharge from pumping wells is simulated in the well package. An adequate zero-flux boundary of the large flow model is defined through testing the sensitivity of the heads at the boundary location to sources and sinks in Guantao. The boundary is sufficiently far away from Guantao, if heads induced by Guantao sources and sinks are close to zero. As a result the following boundary was chosen: In the west the model is bounded by the mountains while it is bounded by the

Yellow River in the east. These boundaries coincide with the natural boundaries of the NCP. The northeast and southwest are chosen approximately parallel to groundwater contour lines (Fig. 8 (a)), which are sufficiently far away to exclude any influence of Guantao sources and sinks. As only the effect of Guantao drivers is of interest in the large model, all sources and sinks outside of Guantao are set equal to zero.

**Figure 8: Discretization map of the large model**

To solve the large model, its parameter distributions are required. The absolute values of parameters outside of Guantao in the large model are not decisive in the study, only the parameter values close to the Guantao boundary are important. The





hydraulic conductivity and specific yield within Guantao have the same values as the calibrated parameters from the Guantao flow model, while the model parameters outside of Guantao are taken from the literature (Liu et al., 2011). The parameter distributions used in the large model are shown in Fig. 9. The sensitivity of the results with respect to the large model parameters will be explored later. The large groundwater flow model is also constructed using VMF-USG. The

modelled area is discretized into 144,829 cells horizontally (Fig. 8). The transmissivity is calculated by multiplying the hydraulic conductivities by the aquifer thickness obtained from the Guantao model in each time step, which means that the transmissivity is constant in each time step but is updated from time step to time step to take into account the changing saturated aquifer thickness. The aquifer thickness outside of Guantao is assumed equal to the average aquifer thickness of the Guantao area. The initial condition is set to zero, which is based on the assumption that the inside drivers of Guantao are in a

quasi-steady-state.  This is consistent with the Guantao flow model, which uses the results from the steady state model as the initial condition in the transient state.

**Figure 9: Parameter distributions map for NCP (modified based on figures in the literature (Liu et al., 2011))**

### 3.2.3 $h_{out}$ flow model

The model for outside contributions is a small scale model with the Guantao administrative boundary as model boundary. The model structure and parameters are defined in the same way as in the Guantao flow model. Since this model is used to determine the groundwater head within Guantao caused by outside drivers, the sources and sinks terms inside are set to zero. The transmissivity used in the model is defined in the same way as in the large model. The specified head values on the Guantao border are obtained by subtracting the head changes determined by the large flow model from the measured heads

on the boundary. Since the initial heads for the large model are zero everywhere, the initial heads for the $h_{out}$ flow model are equal to the initial heads of the Guantao flow model.

## 4 Results and discussion

### 4.1 Groundwater flow characteristics in Guantao

The groundwater contour map calculated from the steady state model is presented in Fig. 10. Two groundwater depression

cones have formed in Guantao, which have a dominant influence on the groundwater flow direction with head gradients from the outside towards the centres of the depression cones. The large-scale regional groundwater flow is directed across the NCP from southwest to northeast. Due to over-pumping, the groundwater level has been declining continuously since the 1960s (Cao et al., 2013). The intensive local pumping caused the development of numerous depression cones in the NCP. Analysis of past data shows that the depression cone in Guantao formed in the early 2000s (Zheng et al., 2010; Shao et al.,

2013; Cao et al., 2013). The groundwater flow characteristics are consistent with the conclusions from former studies. Groundwater level gradients are steeper in the East than in the West, which is caused by the higher observed head values on





the eastern boundary, especially in its south-eastern and north-eastern sections. The groundwater recharge from precipitation and irrigation amounts to 141,000 m³/d, which is over 87% of the total input to Guantao. The net inflow from the specified head boundary accounts for only 12% of the total input. Weiyun River is located on the eastern boundary of Guantao County. Since the Guantao boundary is simulated as a specified head boundary, the infiltration from Weiyun River contributes to the net inflow from the head boundary. The infiltration from the Weixi channel constitutes about 0.9 % of the total inflow. The main sink is due to pumping wells extracting groundwater from the aquifer. It amounts to over 96% of the total outflow. The rest is due to boundary flow directed to the outside.

**Figure 10: Groundwater contour map of the steady state (in m a.s.l.)**

The groundwater balance of the transient model is presented in Fig. 11. The boundary inflow and main channel infiltration are not shown in the figure because they occupy only a small part of the total groundwater recharge. The infiltration from precipitation and irrigation fluctuates gently with time due to the delayed and damped groundwater recharge from precipitation and irrigation backflow through the long soil column as discussed in Sect. 3.2.1. The pumping rate fluctuates strongly in the first six years, while after that its variability decreases. This is due to the temporal behaviour of precipitation, which directly influences the groundwater withdrawals for irrigation in summer (Fig. 2). The aquifer storage increases when groundwater recharge exceeds groundwater withdrawals. This is the case in 2003, where groundwater storage recovers by $19.6 \times 10^6$ m³/a. In the following years, it fluctuates negatively and positively and reaches a smaller value of recovery of 2.7 $\times 10^6$ m³/a at the end of the simulation period. Between 2003 and 2011 the groundwater storage is depleted at a rate of 0.18 $\times 10^6$ m³/a only, which confirms that using the data set of 2003-3011 for the steady state model was justified. The dashed line in Fig. 11 represents the annually averaged groundwater level dynamics in Guantao County. It is obvious that the groundwater head increases with decreasing groundwater withdrawals.

**Figure11:  Annual water budget for the period 2003-2012**

**4.2 Groundwater level changes due to inside and outside drivers**

After running the three corresponding numerical models, the spatial distribution of the groundwater level and its decomposition at the final time step are presented in Fig. 12. The large light green area in Fig. 12(a) with $\Delta h_{\text{Guantao}}$ values around zero indicates that the large model boundary is sufficiently far away from Guantao to make the chosen boundary conditions irrelevant.  $\Delta h_{\text{Guantao}}$ gradually changes to non-zero values around and in Guantao area where inside drivers dominate. Within Guantao the values of $\Delta h_{\text{Guantao}}$ change from -5 m in the south to 4 m in the north (Fig. 12 (b)), which means that the driving forces inside Guantao did not contribute to groundwater head changes ($\Delta h_{\text{Guantao}}$) in the central part of the county. The groundwater depression cone in the south is attributed to the locally intensive groundwater exploitation. The groundwater head contour map in Fig. 12 (c) shows that groundwater in Guantao flows from northeast and southeast to





the groundwater depression cone, which is consistent with the result obtained from the steady state model. The groundwater head gradients are steep in the east while they are flat in the west. The groundwater flow field determined by outside drivers is more uniform than changes superimposed by inside drivers (Fig. 12 (d)). Its flow is from east and south to northwest. The regional groundwater flow directions in Guantao are governed by both inside and outside drivers while the groundwater head

depression cones formed in Guantao are caused by inside drivers.

The groundwater head in Guantao and its decomposition obtained from numerical models are supposed to satisfy Eq. (1). The difference between the $\Delta h_{\text{Guantao}}$ calculated by subtracting Fig. 12 (d) from Fig. 12 (c) according to Eq. (1) and the $\Delta h_{\text{Guantao}}$ obtained from the large flow model is shown in Fig. 12 (e). Without an error it should be zero everywhere.

Apparently, the error is quite large in the over-pumped area. It varies from -1.7 m in the south to 0.8 m in the north. This error is caused by the difference in the types of boundary conditions used in the three numerical models. For the large model, there are no specified heads on the Guantao border. However, if one superimposes two solutions, the boundary has to be superimposed consistently as well. To prove this, another small-scale numerical model within Guantao's administrative boundary is used to update $\Delta h_{\text{Guantao}}$ within Guantao County. This $\Delta h_{\text{Guantao}}$ flow model within Guantao has the same

model structure, parameters and sources and sinks as the Guantao model. The only difference is that the values for $\Delta h_{\text{Guantao}}$ along the specified head boundary are obtained from the large model. After running the $\Delta h_{\text{Guantao}}$ flow model within Guantao with this boundary condition, the new distribution of $\Delta h_{\text{Guantao}}$ is shown in Fig. 12 (f). The distribution of the updated $\Delta h_{\text{Guantao}}$ is the same as shown in Fig. 12 (b) but the absolute values are different, varying from -7 m in the south to 5 m in the north. This shows that the specified head condition on the Guantao border limits the influence of inside

drivers. The difference between the $\Delta h_{\text{Guantao}}$ calculated according to Eq. (1) and the $\Delta h_{\text{Guantao}}$ obtained from the small-scale model is shown in Fig. 12(e). The error is uniformly very small with a maximum of 0.03 m (Fig. 12 (g)). The error distribution in other time steps can also be calculated in the same way. The largest error over the whole time period is around 0.04 m, which is acceptable given the limited accuracy of the numerical solution. This means the correct results are obtained, if we calculate $\Delta h_{\text{Guantao}}$ according to Eq. (1). This method is chosen in all the following discussions.

**Figure 12: The spatial distribution of the groundwater head and its decomposition in Guantao at the final time step. (Fig. 12 (a) refers to the spatial distribution of head changes due to inside drivers calculated by the large model; Fig. 12 (b) is a zoom into the large model showing only Guantao County ($\Delta h_{\text{Guantao}}$); Fig. 12 (c) is the groundwater head map of Guantao obtained from the Guantao model ($h_{\text{Guantao}}$); Fig. 12 (d) is the groundwater head distribution determined by outside drivers and obtained from the**

**$h_{\text{out}}$ flow model ($h_{\text{out}}$); Fig. 12 (e) shows the inconsistency of $\Delta h_{\text{Guantao}}$ in the large model; Fig. 12 (f) refers to the distribution of the updated $\Delta h_{\text{Guantao}}$; Fig. 12 (g) shows the error of the updated $\Delta h_{\text{Guantao}}$)**

The groundwater head in the Guantao flow model can be decomposed at any grid point and at any time. The time series of groundwater heads and their decomposition at three chosen locations are shown in Fig. 13 (two are located within Guantao




and one is on the boundary). The black solid line refers to $h_{\text{Guantao}}$ obtained from the Guantao model, the dashed line represents $h_{\text{out}}$ obtained from the $h_{\text{out}}$ flow model in Guantao and the dotted line is $\Delta h_{\text{Guantao}}$ , which is calculated using Eq. (1). Figure 13 (a) and Fig. 13 (c) show that $h_{\text{out}}$ changes more smoothly in time than $h_{\text{Guantao}}$ and $\Delta h_{\text{Guantao}}$. The dynamics of $h_{\text{Guantao}}$ follow the same pattern as $\Delta h_{\text{Guantao}}$, which means that the temporal groundwater head fluctuations within Guantao are caused by inside drivers. It is obvious that the $h_{\text{out}}$ in the south increases fast at the beginning of the simulation period and later levels off at some value. The fluctuations of $h_{\text{out}}$ on the boundary are comparable to those within Guantao, because the specified boundary heads, containing both the influence from inside and outside drivers, do not fluctuate much (Fig. 13 (b)).

**Figure 13: Time series of groundwater heads in Guantao at three selected locations and their decomposition.**

### 4.3 Inside and outside contributions

The contour colour map of groundwater head changes between the last and the first time steps and its decomposition into inside and outside contributions are shown in Fig. 14. Figure 14 (a) shows that the groundwater heads decrease in the south and increase in the north with $\delta h_{\text{Guantao}}$ varying from -2 m to 1.5 m. A similar trend is also observed in the distribution of $\delta\Delta h_{\text{Guantao}}$ (Fig. 14 (b)). The value of $\delta h_{\text{Guantao}}$ varies from -6 m in the south to 4 m in the north, while the distribution of $\delta h_{\text{out}}$ shows the reverse trend of increasing towards the south and decreasing towards the north with values varying from 5 m to -3 m (Fig. 14 (c)). Therefore the groundwater head changes in Guantao are determined by both inside contributions ( $\delta\Delta h_{\text{Guantao}}$ ) and outside contributions ($\delta h_{\text{out}}$). The inside contribution shows a more pronounced difference between south and north, which is largely compensated by the outside contribution.

**Figure 14: Spatial distribution of changes in groundwater head over the modelled time span and its decomposition.**

To explore the groundwater head change and its decomposition in detail, three specific locations are chosen (Fig. 13). The groundwater head change in the north of Guantao County increases by 0.9 m over the modelled time span, resulting from a decrease in $\delta\Delta h_{\text{Guantao}}$ by 0.2 m and an increase in $\delta h_{\text{out}}$ by 1.1 m (Fig. 13 (a)). This indicates that the outside contribution ($\delta h_{\text{out}}$) dominates the groundwater head changes ($\delta h_{\text{Guantao}}$). $\delta h_{\text{Guantao}}$ in Fig. 13 (b) only increases by 0.02 m with the value of $\delta\Delta h_{\text{Guantao}}$ implying an increase of 0.36 m compensated by $\delta h_{\text{out}}$ with a decrease of 0.34 m. In this case, both inside and outside contributions play an important role in determining the final groundwater head changes in the central part of Guantao County. In the south, $\delta h_{\text{Guantao}}$ is the result of a decrease of 5.5 m from $\delta\Delta h_{\text{Guantao}}$ and an increase of 4.7 m from $\delta h_{\text{out}}$ (Fig. 13 (c)). This further proves that considering the inside contribution only is not sufficient to assess the efficiency of groundwater management in Guantao. In such areas, decomposing groundwater head changes into inside and outside contributions is particularly important. In order to assess the work of Guantao Water Department in a just way, the





head changes due to the inside contribution should be the correct criterion. Its trend will show how the internal gap between pumping and recharge develops distributed in space. Only this quantity can be influenced by measures taken in Guantao.

**4.4 Sensitivity of head change and their decomposition to model calibrated parameters**

The sensitivity of the model to parameter changes is analysed to test its robustness. The parameters, hydraulic conductivities

and specific yields, are perturbed by an increase of 50% in the individual runs, while the recharge infiltration ratios are perturbed by an increase of 20%, respectively. (Parameters include 4 hydraulic conductivity values $K1$ - $K4$ and 4 specific yield values $S_y1$- $S_y4$ for the Guantao model, and 6 more hydraulic conductivity values $K5$ - $K10$ as well as 6 more specific yield values $S_y5$- $S_y10$ for the large model). When hydraulic conductivities and specific yields are perturbed, the groundwater heads on the boundary in the Guantao flow model are not changed while the $\Delta h_{\text{Guantao}}$ on the Guantao administrative

boundary obtained from the large model will be influenced. Thus the $h_{\text{out}}$ values on the boundary in the $h_{\text{out}}$ flow model in Guantao are also changed according to Eq. (1). As stated in Sect. 3.2.1, the recharge infiltration ratios are used to calculate the groundwater recharge with Hydrus 1D. Unlike the hydraulic conductivities and specific yields, the perturbed recharge infiltration ratios change sources within Guantao, which will have an impact on the boundary heads. To make the boundary heads for the Guantao model consistent with the perturbed inside sources, the following procedure is implemented: First

$\Delta h_{\text{Guantao}}$ is computed with the large model using the groundwater recharge from Hydrus 1D and the perturbed recharge infiltration ratios. The new groundwater heads on the Guantao boundary are then calculated based on Eq. (1) considering the $h_{\text{out}}$ obtained in Sect. 4.2. Then the Guantao groundwater model is run using the consistent values for boundary heads and sources to calculate the groundwater heads in Guantao. In the next step, the  $h_{\text{out}}$ flow model in Guantao is run using the consistent values for parameters to calculate the groundwater heads in Guantao determined by outside drivers. Then

$\Delta h_{\text{Guantao}}$ is updated based on Eq. (1). Finally, the groundwater head change and its decomposition according to Eq. (6) to Eq. (9) can be determined.

The model sensitivity to parameters is expressed by the normalised composite sensitivity ($S_{NC}$) which is defined in Eq. (12).

$$S_{NC,i} = \frac{\sqrt{\sum_j^n \left( \left( \frac{Y_j^i - Y_j^{i,0}}{P_{per,i} - P_i} \right) * P_i \right)^2}}{n} \tag{12}$$

$Y_j^i$ is any of the modelled groundwater head changes (referring to $\delta h_{\text{Guantao}}$ , $\delta \Delta h_{\text{Guantao}}$ , $\delta h_{\text{out}}$  in the study) in observation well $j$ over the modelled time span using the perturbed model parameters $P_{\text{per},i}$ (m) ; $Y_j^{i,0}$ is the modelled groundwater head change in observation well $j$ using the unperturbed model parameters $P_i$ ; $n$ is the total number of observations.





The normalised sensitivities of the Guantao head change ($\delta h_{\text{Guantao}}$), the inside contribution ($\delta \Delta h_{\text{Guantao}}$) and the outside contribution ($\delta h_{\text{out}}$) to parameters and recharge infiltration ratios are presented in Fig. 15. The higher values of sensitivity to recharge infiltration ratios indicate that recharge has the strongest influence on the groundwater head changes and their inside contribution. The hydraulic conductivities and specific yields ($K5$ to $K10$ and $S_y5$ to $S_y10$) in the large model exert the same influence on the inside and outside contributions since all the parameters are located outside of Guantao area. The sensitivities of the models to the thickness shown in Fig. 15 are obtained by increasing the aquifer thickness outside of Guantao by a factor of 2. The aquifer thickness outside of Guantao influences the values of transmissivity in the large model therefore the thickness influences both inside and outside contributions and has no influence on the Guantao head changes. The highest sensitivity to specific yields within Guantao ($S_y1$ to $S_y4$) was observed in the inside contribution, followed by the outside contribution and the Guantao head changes. The smallest sensitivity to hydraulic conductivities within Guantao was observed for the outside contribution. $K1$ and $K2$ exert a larger influence on the Guantao head changes ($\delta h_{\text{Guantao}}$) than on the inside contribution ($\delta \Delta h_{\text{Guantao}}$).

**Figure 15: The normalised sensitivity of head change and contributions to parameters and recharge ratios**

## 4.5 Inside and outside contributions affected by the pumping rate

Similar simulations are also carried out regarding the Guantao pumping rates. In a sensitivity test the pumping rate is decreased by 50%. The procedures to guarantee consistency are similar to those used for changing the recharge infiltration ratios (see Sect. 4.4). The spatial distributions of groundwater head changes and their decomposition with the perturbed pumping rate are shown in Fig. 16. The differences in groundwater head changes range from 2 m in the southwest to 13 m in the northeast (Fig. 16 (a)). $\delta h_{\text{Guantao}}$ is around 1 m in the southwest corner and increases to a maximum of 12 m in the northeast. In contrast, the outside contribution $\delta h_{\text{out}}$ is negative in the northeast and increases to 6 m in the southwest.

Two specific locations (one in the south and the other in the north) are chosen to analyze the influences from inside and outside contributions on the groundwater head changes. The $\delta \Delta h_{\text{Guantao}}$ in the north increases by 11 m while $\delta h_{\text{out}}$ only increases by 1.1 m. The two components determine that $\delta h_{\text{Guantao}}$ increases by 12.1 m over the modelled time span. This indicates that inside contributions ($\delta \Delta h_{\text{Guantao}}$) play a much more important role in determining the groundwater head changes ($\delta h_{\text{Guantao}}$). In the south the groundwater head changes show an increase of 6.4 m, of which 4.7 m is from the outside contribution and 1.7 m from the inside contribution, which means that in this region $\delta h_{\text{Guantao}}$ is mainly influenced by the outside contribution.

**Figure 16: Spatial distribution of groundwater head change over the modelled time span and its decomposition: Influence of decreased pumping rate.**

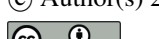



### 4.6 Inside and outside contributions affected by specified head boundary conditions

To explore the impact of the boundary conditions on the groundwater head change and its decomposition, the Guantao model is run with specified heads on the boundary increased by 2 m uniformly. The differences between the models with and without changing the values of the boundary heads are shown in Fig. 17. Positive values indicate an increasing trend in the component compared to the results without increased specified head values. Since there are no changes in sources and sinks in Guantao, the difference in $\delta\Delta h_{\text{Guantao}}$ is around zero everywhere (Fig. 17 (b)). The differences in $\delta h_{\text{Guantao}}$ and $\delta h_{\text{out}}$ are increased by 2 m almost everywhere except on the boundary where the differences are zero (Fig. 17 (a) and Fig. 17 (c)). It is clear that changing specified heads on the boundary will only have a strong impact on the outside contribution. It is also important to keep in mind that the Weiyun River infiltration is implicitly considered in the specified head boundary condition, which is supposed to influence the inside contribution.

**Figure 17: Differences between the model runs with and without increased specified heads on the boundary**

### 5 Conclusions

This study demonstrates how to decompose the groundwater head changes within a political boundary into inside and outside contributions using numerical models. The groundwater head in Guantao was calculated using the Guantao flow model. The groundwater head changes on the Guantao boundary caused by inside drivers were computed in a large model with a boundary far from the Guantao administrative boundary. The difference between values on the Guantao boundary from the two models is assigned as specified head boundary condition for the $h_{\text{out}}$ flow model in Guantao to calculate the groundwater head contribution in Guantao determined by outside drivers. There are no sources and sinks in this model. To eliminate inconsistencies caused by the different types of boundary conditions in the small and big models, the $\Delta h_{\text{Guantao}}$ within Guantao can be either calculated according to Eq. (1) or recomputed by using a small scale groundwater flow model with the Guantao administrative boundary as prescribed head boundary with boundary values for $\Delta h_{\text{Guantao}}$ obtained from the large flow model.

The steady state model of Guantao was used to calibrate hydraulic conductivities and recharge infiltration ratios using data averaged over the 10-year time interval from 2003 to 2012. All hydraulic conductivities and recharge ratios are sensitive to head observations, with recharge infiltration ratios having higher sensitivities than hydraulic conductivities. The identified recharge infiltration ratios are correlated with each other with correlation coefficients around -0.6. The calibrated hydraulic conductivities and recharge infiltration ratios are kept fixed in the transient model and four zonal specific yield values are also calibrated. Hydrus 1D is used to generate the groundwater recharge at the groundwater table since the soil column leads



to considerable delay and damping of the seepage at the surface. The calibrated groundwater heads fit observations quite well with an $E_{\mathrm{RMS}}$ of less than 2 m.

Based on numerical models discussed in the study, the groundwater head in Guantao at any time and at any point can be decomposed into the groundwater head changes determined by inside drivers and the groundwater head determined by outside drivers. The groundwater head change over the whole modelled time span can correspondingly be decomposed. The groundwater head in Guantao decreased by -2 m to 1.5 m. The inside contribution induced a decrease by -6 m to 4 m, while the outside contribution caused a variation between a decrease of 3 m and an increase of 5 m. The groundwater head change and its decomposition in three specific locations show that the groundwater head change in the north is impacted mainly by the outside contribution while in the centre and south both inside and outside contributions are determining the development of groundwater head changes. This proves that a separation of inside and outside contributions to groundwater head changes is feasible. Only by separation of contributions one can prove whether the groundwater control efforts in Guantao are successful or not. It could well be that Guantao's management is functioning, but due to the outside contribution, the observed groundwater head changes are still negative.

The sensitivity of groundwater head change and its decomposition to various model parameters was analysed by perturbing parameters by 50% and 20%. The results show that model outputs are sensitive to hydraulic conductivities, specific yields and recharge infiltration ratios within Guantao itself. Parameters outside of Guantao have an influence on the large model and the $h_{\mathrm{out}}$ flow model only. The recharge infiltration ratios have a stronger influence on the groundwater head change ($\delta h_{\mathrm{Guantao}}$) and the inside contribution ($\delta \Delta h_{\mathrm{Guantao}}$) than hydraulic conductivities and specific yields. Guantao shallow aquifer is currently not or only very slightly over-pumped. It is feasible, to get the aquifer into equilibrium at present groundwater levels.

*Author contributions.* NL conducted the model simulations. NL and WK performed the analysis and wrote the manuscript. HL and WL contributed to the data, results analysis and model setup. FC contributed to the data and model setup.

*Competing interests.* The authors declare that they have no conflict of interest.

*Acknowledgements.* This research was supported financially by the Swiss Agency for Development and Cooperation (SDC) under the project "Rehabilitation and management strategy for over-pumped aquifers under a changing climate". We thank Mr. Junfeng Gu and Mr. Hongliang Liu (Handan Department of Water Resources) and Mr. Huaixian Yao, Mr. Fei Gao and Mr. Guangchao Li (Guantao Department of Water Resources) for the tremendous effort they put into our data collection.



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





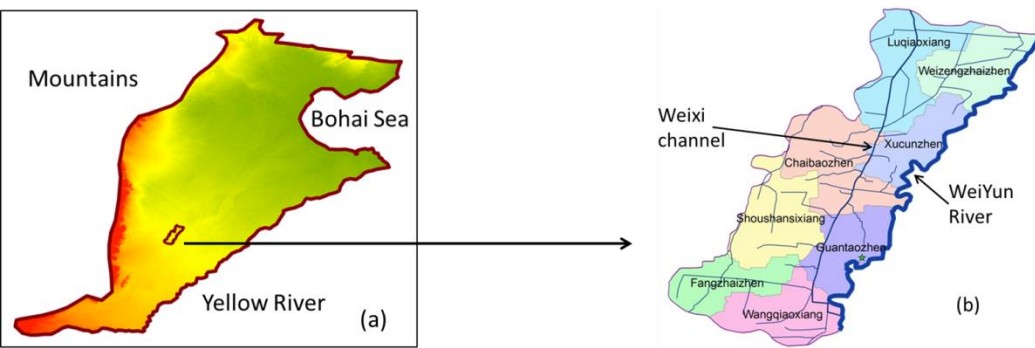

**Figure 1:**

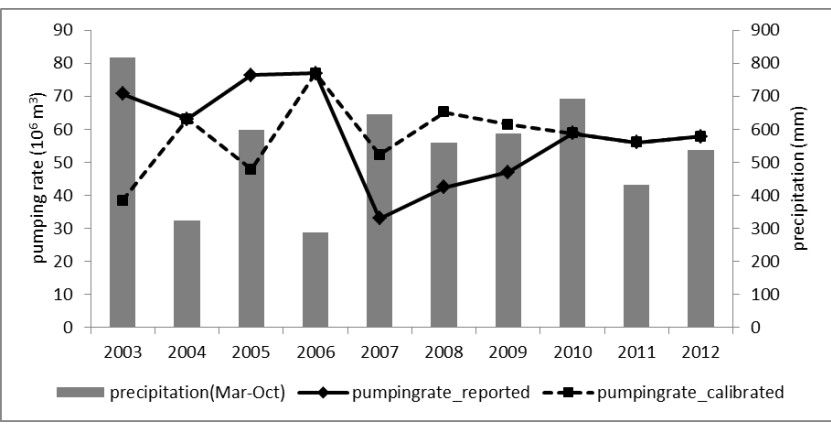

5    **Figure 2:**

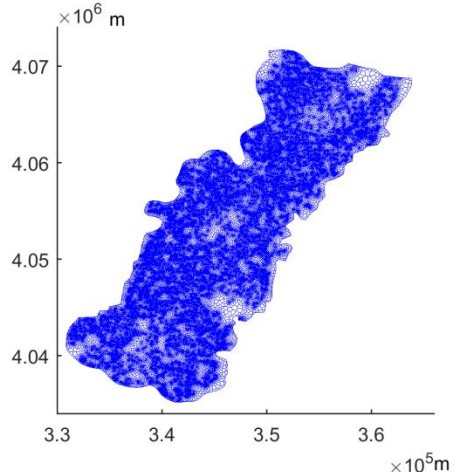

**Figure 3:**





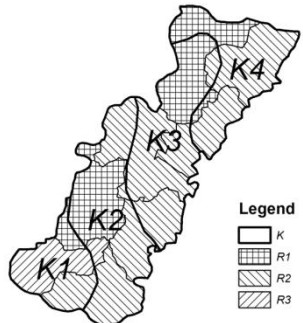

**Figure 4:**

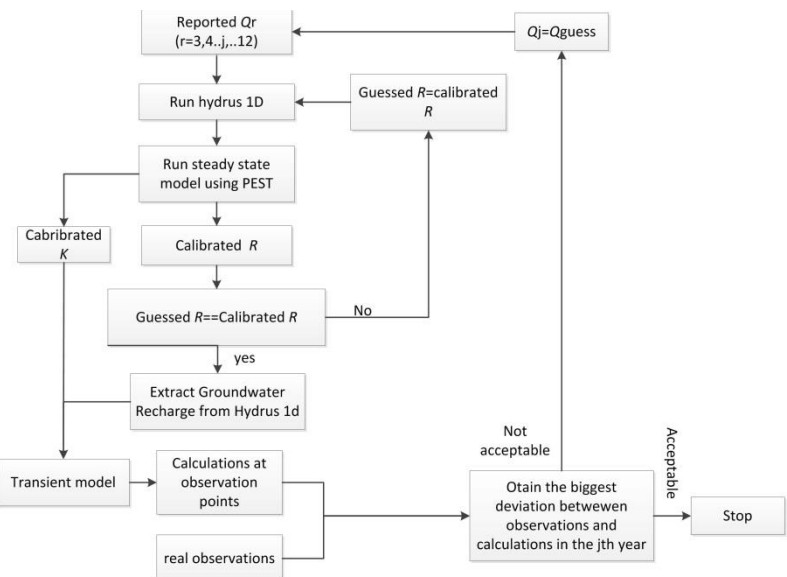

**Figure 5:**

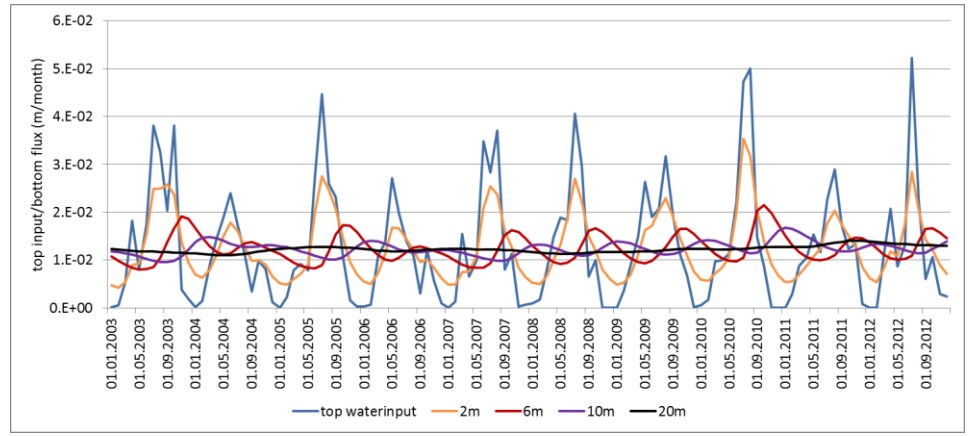

**Figure 6:**




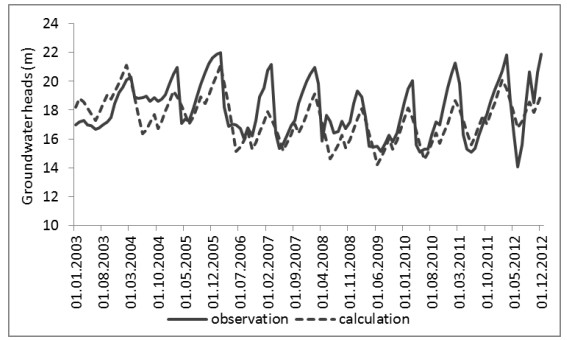
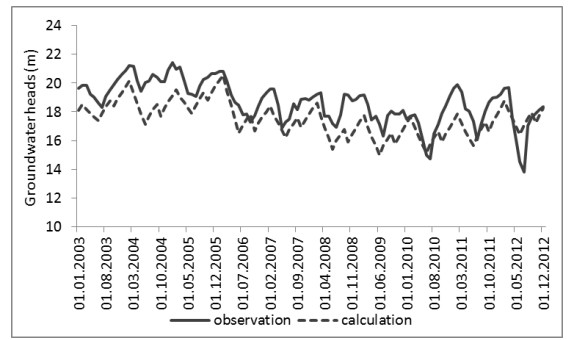
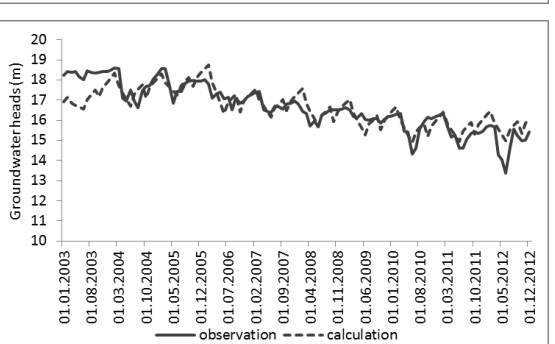
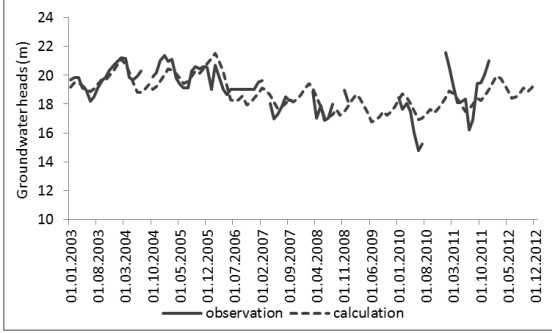

**Figure 7:**

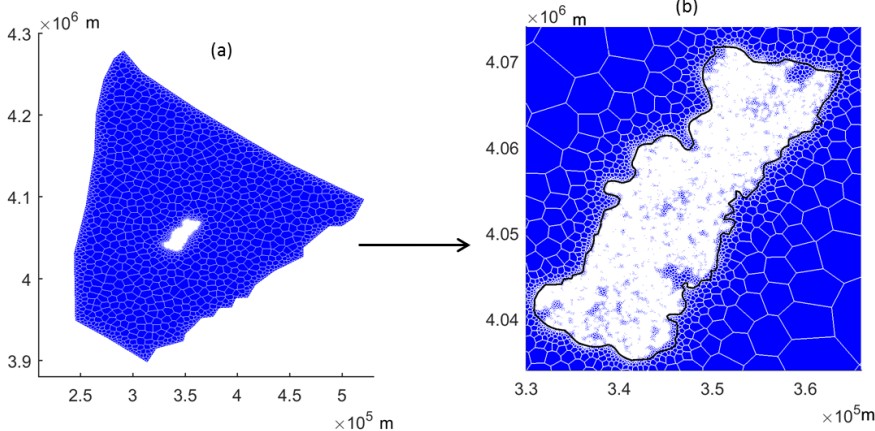

5  **Figure 8:**





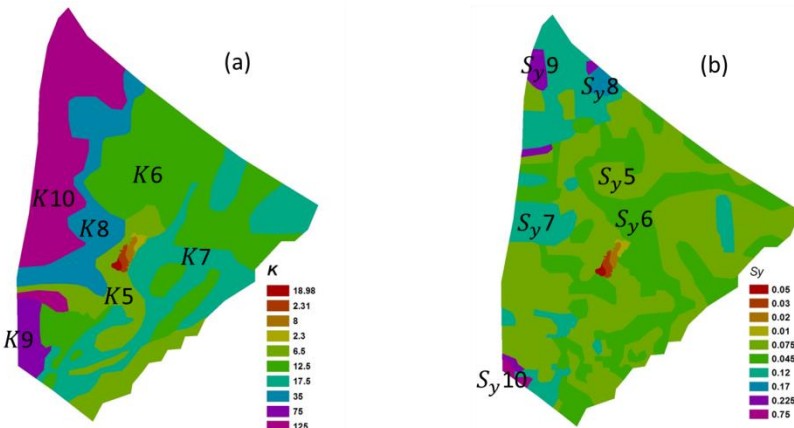

**Figure 9:**

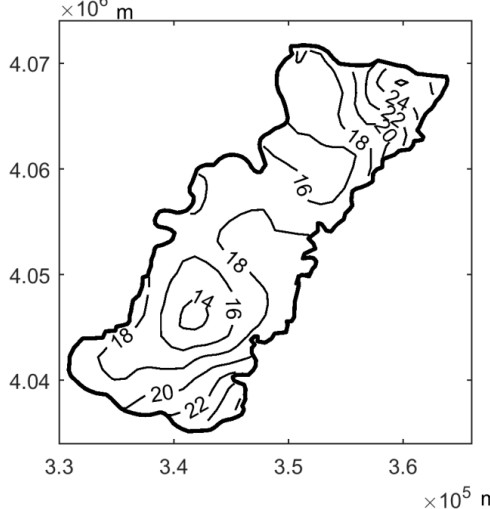

5   **Figure 10:**

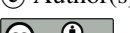



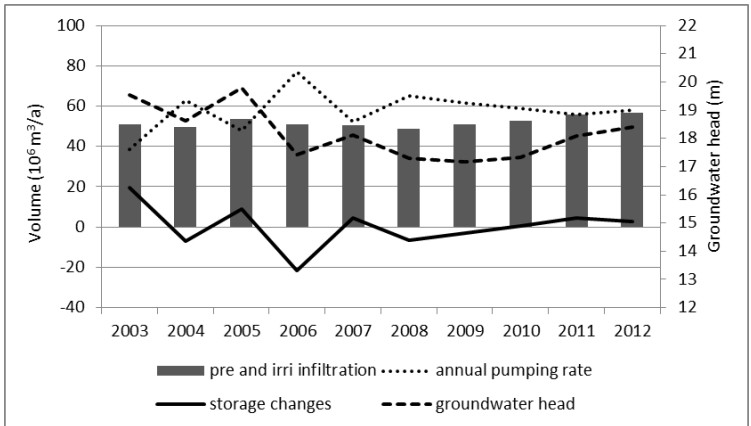

**Figure 11:**

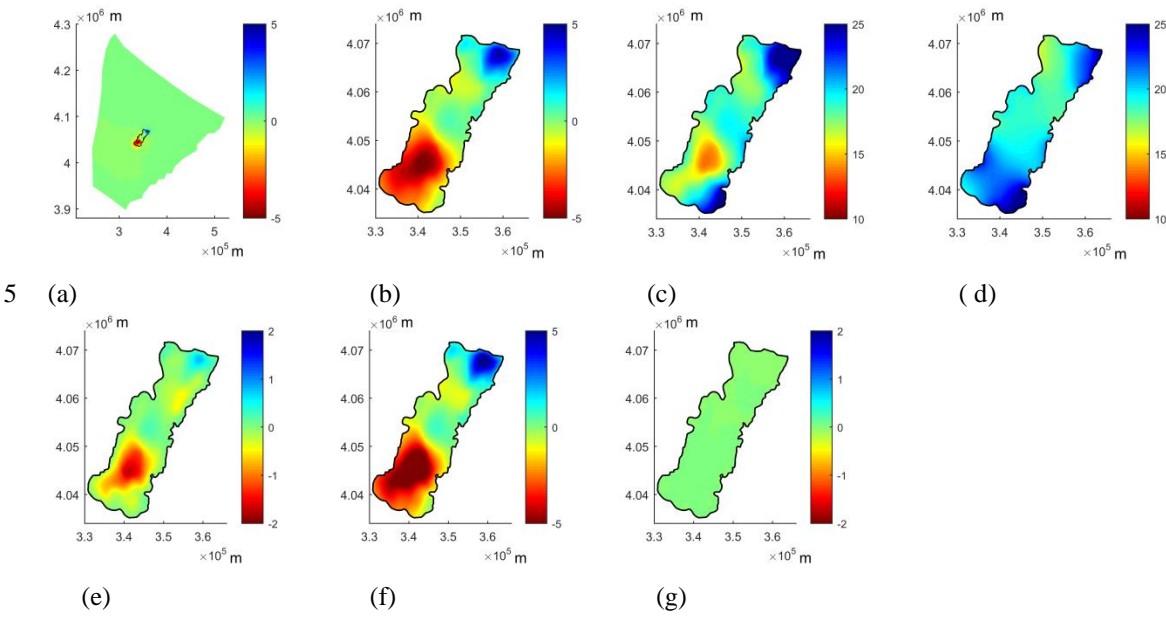

5   (a)                        (b)                        (c)                              ( d)

    (e)                        (f)                        (g)

**Figure 12:**



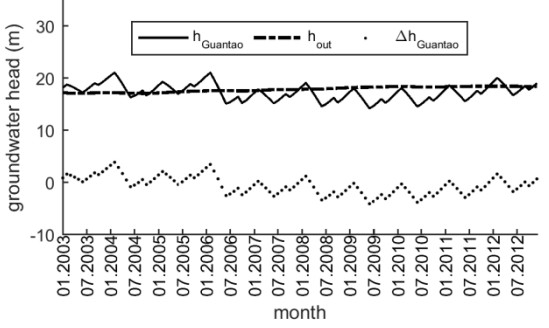

**(a)  Location in the north**

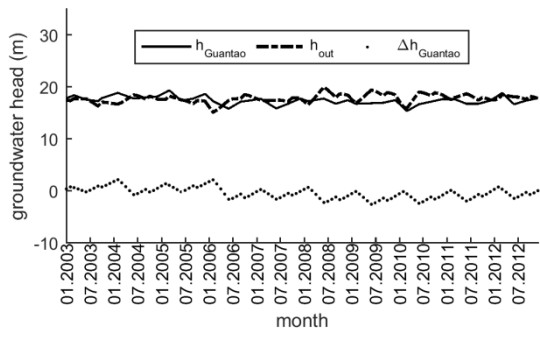

**(b)  Location in the middle on the boundary**

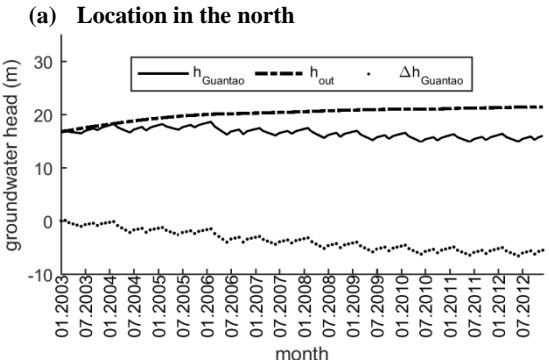

**(c)  Location in the south**

**Figure 13:**

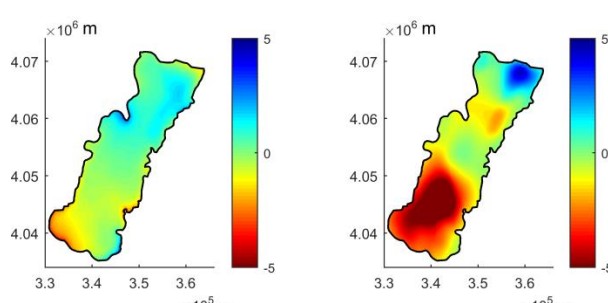

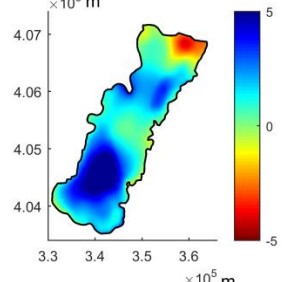

**(a)  Groundwater   head   changes   over time ($\delta h_{\text{Guantao}}$)**

**(b) Inside contribution ( $\delta \Delta h_{\text{Guantao}}$ )**

**(c) Outside contribution  ($\delta h_{\text{out}}$)**

**Figure 14:**





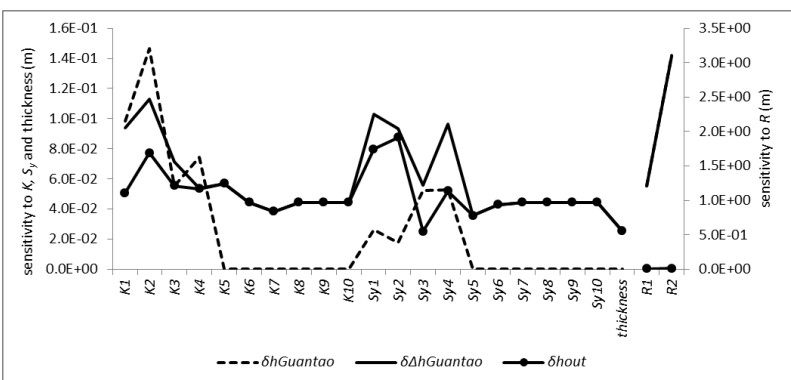

**Figure 15:**

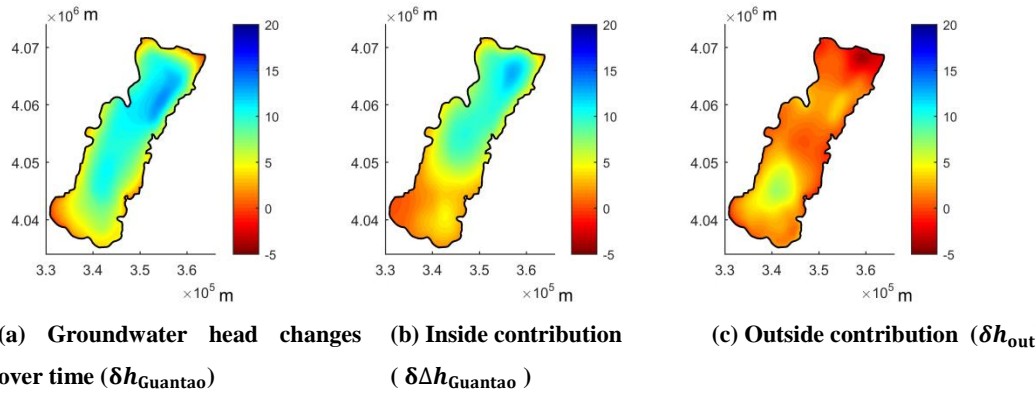

(a) Groundwater head changes over time ($\delta h_{\mathrm{Guantao}}$)  (b) Inside contribution ($\delta\Delta h_{\mathrm{Guantao}}$)  (c) Outside contribution ($\delta h_{\mathrm{out}}$)

5  **Figure 16:**

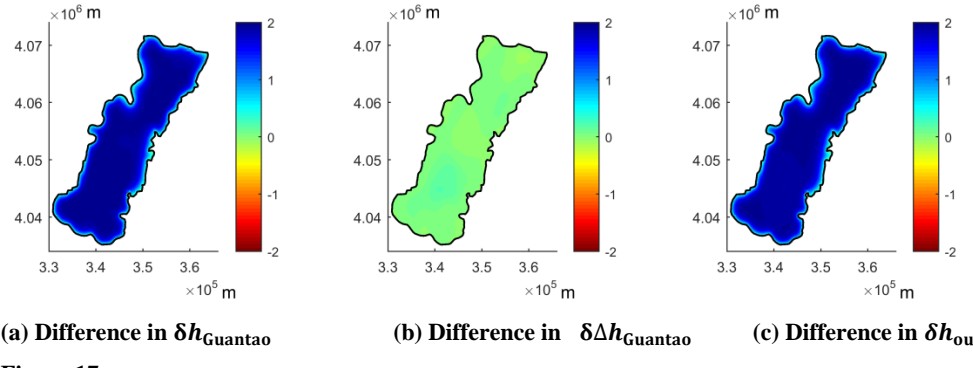

(a) Difference in $\delta h_{\mathrm{Guantao}}$  (b) Difference in $\delta\Delta h_{\mathrm{Guantao}}$  (c) Difference in $\delta h_{\mathrm{out}}$

**Figure 17:**



Table 1. The automatically calibrated values of hydraulic conductivities and recharge infiltration ratios and their respective sensitivity to observed heads

| Parameter | Values (m/d or-) | 95% percent confidence limits | | Sensitivity |
| --- | --- | --- | --- | --- |
| | | lower limit | upper limit | |
| K1 | 11.7 | 7.9 | 17.3 | 0.42 |
| K2 | 4 | 2.4 | 6.6 | 0.50 |
| K3 | 8 | 3 | 21 | 0.20 |
| K4 | 1.7 | 0.8 | 3.6 | 0.26 |
| R1 | 0.16 | 0.12 | 0.2 | 1.35 |
| R2 | 0.21 | 0.19 | 0.22 | 4.13 |

5  Table 2. Correlation coefficients among different parameters

| | K 1 | K 2 | K 3 | K 4 | R1 | R2 |
| --- | --- | --- | --- | --- | --- | --- |
| K1 | 1.00 | -0.28 | 0.02 | 0.10 | -0.05 | -0.06 |
| K2 | | 1.00 | -0.58 | 0.36 | -0.49 | -0.11 |
| K3 | | | 1.00 | -0.10 | 0.13 | 0.31 |
| K4 | | | | 1.00 | -0.36 | -0.25 |
| R1 | | | | | 1.00 | -0.57 |
| R2 | | | | | | 1.00 |