# Peer review of "Decomposition technique for contributions to groundwater heads from inside and outside of an arbitrary boundary: Application to Guantao County, North China Plain"

_Hydrology and Earth System Sciences, 2018_

## Referee Comment (RC1) · Anonymous Referee #1 · 9 Feb 2019

Comments to "Decomposition technique for contributions to groundwater heads from inside and outside of an arbitrary boundary: Application to Guantao County, North China Plain" submitted to HESS.

**General comments:**

This paper presents a decomposition technique to describe the contributions to groundwater heads from inside and outside drivers. This technique could provide useful information for groundwater management of an administrative unit. Guantao County of Hebei Province, China, serves as an example to demonstrate the decomposition technique, and this technique is implemented by building three groundwater models. This manuscript is well organized and easy to read.

Building reliable groundwater models is crucial to this technique, and the results depends on accurate groundwater models. I have some questions on these groundwater models (e.g., calibration, model parameters) and the results of this paper, please see the specific comments below. Therefore, these questions should be clarified before publication, and a moderate revision is recommended to this paper.

**Specific comments:**

1. As for the unsaturated zone of study area with a thickness of 20m, although there is no phreatic evaporation from the aquifer, the evapotranspiration process of recharge to groundwater (e.g., precipitation or irrigation) is not neglectable, because some precipitation can't contribute to effective groundwater recharge due to this thick unsaturated zone. As stated at line 23-24 of page 9, "The water input for the groundwater surface equals precipitation plus irrigation minus evaporation and evapotranspiration," and how to evaluate this process.

2. The amount of groundwater pumping $Q$ is an important input for groundwater model, especially when the observed water head of the pumping well is used for model

calibration. Is this parameter $Q$ calibrated by using PEST? It would be nice to give a summarized information about the calibrated model parameters and their prescribed ranges.

3. According to the line 30-32 of page 9, the PEST is used to calibrated the parameters of steady state model, and then these parameters are used for the transient model, and the parameter specific yield is adjusted manually. I think the calibrated parameters from steady state model used for transient model may be problematic, because the two models have different input-output relationship. Why not calibrate the transient model by PEST?

4. Line 17 of page 12. The transient model has four specific yield variables, are these variables have fixed proportional relationship? otherwise, it is difficult to adjust these variables manually.

5. Line 23-27 of page 12, and the first half of page 13, the fitting results are used to represent the performance of model calibration. However, I think the data could be divided into two parts, one for model calibration, and the other for model validation.

6. What's the definition of the "sensitivity" in Table 1, or how to calculate it in this paper?

7. It would be nice to quantify the contributions of inside and outside drivers to the overall groundwater flow field of Guantao county, such as the **% average variation of Guantao's groundwater flow field is contributed by the inside drives, …, because this information is of interest to the manager of local water resources administration.

---

## Author Comment (AC1) · 9 Mar 2019

Comments to "Decomposition technique for contributions to groundwater heads from inside and outside of an arbitrary boundary: Application to Guantao County, North China Plain" submitted to HESS.

General comments: This paper presents a decomposition technique to describe the contributions to groundwater heads from inside and outside drivers. This technique could provide useful information for groundwater management of an administrative unit.

[Figure]

Guantao County of Hebei Province, China, serves as an example to demonstrate the decomposition technique, and this technique is implemented by building three groundwater models. This manuscript is well organized and easy to read.

Building reliable groundwater models is crucial to this technique, and the results depends on accurate groundwater models. I have some questions on these groundwater models (e.g., calibration, model parameters) and the results of this paper, please see the specific comments below. Therefore, these questions should be clarified before publication, and a moderate revision is recommended to this paper.

Specific comments: 1. As for the unsaturated zone of study area with a thickness of 20m, although there is no phreatic evaporation from the aquifer, the evapotranspiration process of recharge to groundwater (e.g., precipitation or irrigation) is not neglectable, because some precipitation can't contribute to effective groundwater recharge due to this thick unsaturated zone. As stated at line 23-24 of page 9, "The water input for the groundwater surface equals precipitation plus irrigation minus evaporation and evapotranspiration," and how to evaluate this process.

1.1 Response

We do not simulate the soil water processes. Our recharge simulation starts at the depth of the soil water shed. The amount of water arriving at this depth is the total water input by irrigation and precipitation minus all losses by evapotranspiration, which are assumed to be a fixed percentage of total input. The simulation of the soil column down to the water table is necessary, as given the large depths to groundwater of more than 20 m, there is a considerable delay and temporal redistribution of the net input. It is the flux at the water table which – together with abstraction by pumping - is responsible for the groundwater level dynamics.

1.2 Modification

The paragraph on page 9 is changed as follows:" The groundwater recharge is ob-
tained from the calibrated steady state model as a percentage of total precipitation plus irrigation, which enters the soil column below the root zone. Monthly inputs are computed according to rainfall and irrigation events with this constant recharge ratio. Due to the large depth to groundwater of 20 m and more the input is delayed and attenuated. The final temporal distribution of the flux at the groundwater table is calculated in the Hydrus simulation. The upper boundary of the column is implemented as an atmospheric boundary while a free drainage condition is implemented at the lower boundary. . . .."

2. The amount of groundwater pumping Q is an important input for groundwater model, especially when the observed water head of the pumping well is used for model calibration. Is this parameter Q calibrated by using PEST? It would be nice to give a summarized information about the calibrated model parameters and their prescribed ranges.

2.1 Response

Generally the pumping rates are among the collected input data for the groundwater model. After analyzing the available time series of collected pumping rates, several yearly data are questioned. The reason is shown in the first paragraph on page 5 "The pumping rate for irrigation is highly dependent on precipitation in the NCP. Pumping rates are generally less in years with higher precipitation. Hence, the data point for 2003 showing a combination of higher precipitation with a larger pumping rate is questionable. The sudden significant decrease in the reported pumping rate after 2006 is also questionable as there were hardly any changes in cropping area, crop types and irrigation methods." Therefore, the pumping rates for these years are adjusted during the manual calibration of the transient state model. PEST in our case is only used to calibrate the steady state model.

2.2 Modification

The paragraph on page 5 is changed as "Therefore, the pumping rates for these years

will be adjusted during manual calibration of the transient numerical model."

3. According to the line 30-32 of page 9, the PEST is used to calibrated the parameters of steady state model, and then these parameters are used for the transient model, and the parameter specific yield is adjusted manually. I think the calibrated parameters from steady state model used for transient model may be problematic, because the two models have different input-output relationship. Why not calibrate the transient model by PEST?

3.1 Response

The input and output items used for the two models are the same. The transient model uses the time series data between 2003 to 2012 while the steady state model uses the averages of inputs and outputs in the period from 2003 to 2011. This is feasible as the head averaged over all observations at the end of 2002 is almost identical to that at the end of 2011, implying an average steady state over that period. The steady state model describes the average behavior over the whole period while the transient model describes the transient behavior within that period. As the model is basically linear, the input-output relationships of the two models should be the same. I agree that one can calibrate all parameters in the transient model using PEST.

3.2 Modification

No modification.

4. Line 17 of page 12. The transient model has four specific yield variables, are these variables have fixed proportional relationship? otherwise, it is difficult to adjust these variables manually.

4.1 Response

These variables have no fixed proportional relationship. The manual adjustment is based on the observed groundwater level amplitudes in the respective zones and their absolute calibrated values contain some subjectivity.

4.2 Modification

Changes are made on page12:" The manually calibrated values are to some extent subjective."

5. Line 23-27 of page 12, and the first half of page 13, the fitting results are used to represent the performance of model calibration. However, I think the data could be divided into two parts, one for model calibration, and the other for model validation.

5.1 Response

We agree that you could divide the data into two parts to calibrate and validate (split sampling). If one can fit the whole time series well with one set of parameters obtained by using all data available, that set will also perform well in a split sample procedure.

5.2 Modification

No modification.

6. What's the definition of the "sensitivity" in Table 1, or how to calculate it in this paper?

6.1 Response

The parameter sensitivities are obtained by running PEST. They are stored in the file *.sen. The sensitivity here refers to the composite parameter sensitivity in PEST which is defined as $S(c,i)=sqrt((J^t QJ)ii )/n$ Where J is the Jacobian matrix, Q is the weight matrix, n is the number of observations, i is the ith parameter.

6.2 Modification

Changes are made on page 11: "The parameter (composite) sensitivities are automatically computed by PEST and saved in the file *.sen . They are calculated according to the weighted Jacobian matrix and the number of observations (Doherty, 2003)."

7. It would be nice to quantify the contributions of inside and outside drivers to the overall groundwater flow field of Guantao county, such as the **% average variation of

Guantao's groundwater flow field is contributed by the inside drives, . . ., because this information is of interest to the manager of local water resources administration.

7.1 Response

We totally agree.

7.2 Modification

Changes are made on page 17: "In brief, the groundwater head changes over time in the whole of Guantao County is around 0.12 m, of which -1.9 m is caused by inside contributions and +2.06 m is due to outside contributions. This amounts to an aggregated change of groundwater head over time of 3.96 m, of which 48% are contributed by inside drivers, while 52% are contributed by outside drivers. "

The information is also added in the abstract and conclusion part.

---

## Referee Comment (RC2) · Anonymous Referee #1 · 14 Mar 2019

I think the authors have answered most of my questions, and I agree with the publication of this manuscript.

---

## Referee Comment (RC3) · Anonymous Referee #2 · 4 Apr 2019

This is an interesting study, focusing on splitting inside and outside contributions to groundwater head changes. The studied issue is commonly encountered by groundwater modelers. I believe this work is helpful for large-scale groundwater simulation study. A minor revision is suggested by the reviewer.

1ïïjŐThe detailed boundary conditions, especially the upper boundary (irrigation, evaporation and evapotranspiration), should be presented when running Hydrus 1D. We recommend a soil water balance study in the NCP (Hu, X., Shi, L., Zeng, J., Yang, J., Zha, Y., Yao, Y., & Cao, G. (2016). Estimation of actual irrigation amount and its

impact on groundwater depletion: A case study in the Hebei Plain, China. Journal of Hydrology, 543, 433-449). The authors may refer to some of balance components from this paper. Four years of tracer experiment data were also given in this paper. I also recommend few other papers for the soil moisture dynamics simulation in the NCP to refine the Hydrus 1D simulation. (Li, X., Zhao, Y., Xiao, W., Yang, M., Shen, Y., & Min, L. (2017). Soil moisture dynamics and implications for irrigation of farmland with a deep groundwater table. Agricultural Water Management, 192, 138-148.; Min, L., Shen, Y., Pei, H., & Jing, B. (2017). Characterising deep vadose zone water movement and solute transport under typical irrigated cropland in the North China Plain. Hydrological Processes, 31(7), 1498-1509.)

2. It will be more convincing to add a discussion or cite reference to support the way of handling channel infiltration, river infiltration, and infiltration from these smaller sub-channels in the manuscript. Some details may be provided to increase the validity of model development.

3. Moreover, the determination of the sources and sinks inside and outside Guantao is critical for the decomposition. The spatial location and the number of pumping wells are highly unknown in this area since there is no official statistical data on these. Under this condition, it may be more reasonable to use to areal sink/source terms (area-averaged discharge) while not using point sink/source terms. Due to importance of sources and sinks during the simulation, a discussion on the modeling uncertainty is required.

4. Some minor comments are provided: 4.1 There are far too many figures (17 figures) in this manuscript. It is better to remove some less relevant ones. 4.2 Is there any groundwater abstraction for domestic use and industrial use in Guantao? I notice that except March, April, May, June, and October, there is no groundwater abstraction in other months during the simulation.

---

## Author Comment (AC2) · 17 Apr 2019

This is an interesting study, focusing on splitting inside and outside contributions to groundwater head changes. The studied issue is commonly encountered by groundwater modelers. I believe this work is helpful for large-scale groundwater simulation study. A minor revision is suggested by the reviewer.

1.ЁİThe detailed boundary conditions, especially the upper boundary (irrigation, evaporation and evapotranspiration), should be presented when running Hydrus 1D. We

recommend a soil water balance study in the NCP (Hu, X., Shi, L., Zeng, J., Yang, J., Zha, Y., Yao, Y., & Cao, G. (2016). Estimation of actual irrigation amount and its impact on groundwater depletion: A case study in the Hebei Plain, China. Journal of Hydrology, 543, 433-449). The authors may refer to some of balance components from this paper. Four years of tracer experiment data were also given in this paper. I also recommend few other papers for the soil moisture dynamics simulation in the NCP to refiне the Hydrus 1D simulation. (Li, X., Zhao, Y., Xiao, W., Yang, M., Shen, Y., & Min, L.(2017). Soil moisture dynamics and implications for irrigation of farmland with a deep groundwater table. Agricultural Water Management, 192, 138-148.; Min, L., Shen, Y., Pei, H., & Jing, B. (2017). Characterising deep vadose zone water movement and solute transport under typical irrigated cropland in the North China Plain. Hydrological Processes, 31(7), 1498-1509.)

1.1 Response

We agree. All mentioned papers are cited in this manuscript.

1.2 Modification

The Paragraphs on page 11 and page 12 are changed to "Tan (2014) shows that in the NCP the average recharge ratio from tracer experiments is 0.16 ranging from 0.11 to 0.21, which is quite close to the results from another tracer test done in NCP by Hu et al. (2016)." and "The response time becomes longer with increasing depth to groundwater while the peak flux becomes smaller with a more even distribution over the year. Similar results can be observed in other areas of the NCP (Li et al., 2017; Min et al., 2017)."

2. It will be more convincing to add a discussion or cite reference to support the way of handling channel infiltration, river infiltration, and infiltration from these smaller sub-channels in the manuscript. Some details may be provided to increase the validity of model development.
**2.1 Response**

We agree. New citations are added.

**2.2 Modification**

The paragraph on page 7 is changed to "This is a common way to deal with the leakage to groundwater from numerous field channels in the irrigated cropland which can also be found in other similar studies (Rejani et al., 2008; Cao et al., 2013)."

3. Moreover, the determination of the sources and sinks inside and outside Guantao is critical for the decomposition. The spatial location and the number of pumping wells are highly unknown in this area since there is no official statistical data on these. Under this condition, it may be more reasonable to use to areal sink/source terms (area-averaged discharge) while not using point sink/source terms. Due to importance of sources and sinks during the simulation, a discussion on the modeling uncertainty is required.

**3.1 Response**

The reason we use point sink/source terms to simulate the pumping rate is due to the investigation conducted in 2012 by Guantao Water Resources Department recording all pumping well locations in Guantao.

We agree that an uncertainty discussion would perfectly complement this manuscript. The sections from 4.4 to 4.6 are sensitivity analyses, which are actually based on one ensemble of model realizations. Another paper mainly focusing on model uncertainty and using the Ensemble Kalman Filter is in preparation and will be submitted in a month's time. In section 4.5 of this manuscript, we will add comparisons between results from models with reduced pumping rate to those without any change. This comparison will reflect the uncertainty of the source/sink term's influence to some degree.

**3.2 Modification**

The paragraph on page 19 is modified to "After increasing the pumping rate by 20%, the groundwater head changes over time averaged over the whole of Guantao County is around - 3.32 m ($\delta$h_Guantao), of which -5.37m is caused by inside contributions ($\delta\Delta$h_Guantao ) while +2.05 m is due to outside contributions ($\delta$h_out). Compared to the results from Sect. 4.3, the increased pumping rate increased the inside contribution from 48.5% to 72.4% when compared to the aggregated groundwater head changes".

The information is also added in the conclusion part.

4. Some minor comments are provided: 4.1 There are far too many figures (17 figures) in this manuscript. It is better to remove some less relevant ones. 4.2 Is there any groundwater abstraction for domestic use and industrial use in Guantao? I notice that except March, April, May, June, and October, there is no groundwater abstraction in other months during the simulation.

4.1.1 Response

We agree.

4.1.2 Modification

Figure 3 and Figure 5 are deleted.

4.2.1 Response

The groundwater abstraction for domestic use and industrial use in Guantao is from the deep aquifer, which has a suitable water quality. The water abstraction from the more mineralized shallow aquifer is used for irrigation only.

4.2.2 Modification

The paragraph on page 4 is modified to "It is discharged through pumping wells for irrigation. The deep layer receives only little recharge from upstream at the piedmont, so groundwater levels have been decreasing since pumping for domestic, industrial and irrigation uses started ."

---

## Author Comment (AC3) · 24 Apr 2019

We thank reviewers for their valuable comments and suggestions on improving this manuscript.
* * *
Creative Commons CC BY license logo